# Forecasting the impact of population ageing on tuberculosis incidence

**Chu-Chang Ku** *[◉], **Peter J. Dodd**[◉]

School of Health and Related Research, University of Sheffield, Sheffield, England, United Kingdom

[◉] These authors contributed equally to this work.
* cku1@sheffield.ac.uk

## Abstract

### Background

Tuberculosis (TB) disease reactivates from distant latent infection or recent (re)infection. Progression risks increase with age. Across the World Health Organisation Western Pacific region, many populations are ageing and have the highest per capita TB incidence rates in older age groups. However, methods for analysing age-specific TB incidence and forecasting epidemic trends while accounting for demographic change remain limited.

### Methods

We applied the Lee-Carter models, which were originally developed for mortality modelling, to model the temporal trends in age-specific TB incidence data from 2005 to 2018 in Taiwan. Females and males were modelled separately. We combined our demographic forecasts, and age-specific TB incidence forecasts to project TB incidence until 2035. We compared TB incidence projections with demography fixed in 2018 to projections accounting for demographic change.

### Results

Our models quantified increasing incidence rates with age and declining temporal trends. By 2035, the forecast suggests that the TB incidence rate in Taiwan will decrease by 54% (95% Prediction Interval (PI): 45%-59%) compared to 2015, while most age-specific incidence rates will reduce by more than 60%. In 2035, adults aged 65 and above will make up 78% of incident TB cases. Forecast TB incidence in 2035 accounting for demographic change will be 39% (95% PI: 36%-42%) higher than without population ageing.

### Conclusions

Age-specific incidence forecasts coupled with demographic forecasts can inform the impact of population ageing on TB epidemics. The TB control programme in Taiwan should develop plans specific to older age groups and their care needs.

**Data Availability Statement:** All analysis code and data are available from the github repository: https://github.com/TimeWz667/AgeingTB.

**Funding:** The study was supported by the Medical Research Council UK (grant MR/P022081/1 to PJD). The funders had no role in study design, data

collection and analysis, decision to publish, or
preparation of the manuscript.

**Competing interests:** A potential conflict of interest
and practical consideration is that co-author Peter
Dodd is also a PLOS ONE handling editor for this
collection. This does not alter our adherence to
PLOS ONE policies on sharing data and materials.

## Introduction

In 2018, tuberculosis (TB) was still the top infectious killer in the world [1]. The End TB strategy aims at a 90% reduction in TB incidence rate by 2035 compared with 2015, but the current global rate of decline of around 2% per year is not on track to achieve this [2]. Latent TB infection risk accumulates over lifetimes while TB transmission is ongoing. The prevalence of latent TB infection is highest in older age groups [3], who not only have had the longest exposure, but were often exposed to higher TB transmission rates in the past. ageing, with associated higher rates of progression [4], thus acts as a demographic driver towards higher per capita TB incidence [5]. In the Western Pacific region, many countries have their highest per capita TB incidence rates among older age groups [1]. Among Western Pacific region countries, China, Hong Kong (China), Japan, Korean, Singapore, and Taiwan are facing both high TB burden and population ageing[6,7].

The age profile of future TB incidence is critical for forecasting public health needs and rational policy design [8]. First, older populations will have higher TB (and background) mortality rates [9,10], which implies added difficulty in meeting treatment success targets. Secondly, older adults have more comorbidities and more complex health care needs, which may lead to a longer care-seeking process and higher healthcare expenditure per case. For instance, patients with chronic lung diseases may have signs or symptoms overlapping with TB, making correctly diagnosing their TB slower and more costly [11]. Thirdly, the proportion of TB cases in older age groups should inform policy making, for example suggesting integrating TB care entry points into long-term care programmes, or through clinician training highlighting older people as a TB risk group with their own diagnostic and management challenges [11].

Quantitatively forecasting the TB incidence age profile needs combined models forecasting demographic change and statistical forecasts of age-specific TB incidence. However, a time series analysis producing age-specific forecasts of the TB incidence has not been published to our knowledge. Use of autoregressive integrated moving average models, often including the seasonality of TB incidence is more common [12], and comoving time series analysis has been applied [13] without age-specific information. Age-specific TB incidence modelling, including the use of age-period-cohort models, has been undertaken but without producing epidemic forecasts (e.g. Iqbal et al. [14] and Wu et al. [15]). Mechanistic mathematical modelling, with age structure, also has the potential to generate forecasts [5,16–18]. Indeed, Arregui et al. [18] developed forecasts for the effects of demographic change on TB epidemics, focussing on four relatively young countries; our interests are in developing statistically rigorous time-series approaches and in focusing on an example of a much older population.

In many settings, the demographic transition and population ageing are outpacing declines in TB incidence, so methods to understand and forecast the impact of changing demography on TB epidemics are needed. We, therefore, developed a statistical method capturing age-specific incidence trends and forecasting future epidemics while accounting for demographic change so more methods to understand and forecast the impact of changing demography on TB epidemics are needed.

## Materials and methods

### Setting and data sources

TB incidence in Taiwan has steadily declined from 64 confirmed TB cases per 100,000 in 2007 to 41 per 100,000 in 2017. Since 2005, the proportion of TB cases in Taiwan over 65 years of age has been over 50% and increasing. Between 2007 and 2017, the average age in Taiwan increased from 36 to 40, and the proportions of adults above 65 rose from 10% to 14% [7].

Notification data of culture-confirmed TB cases, excluding foreigners, were obtained from the Taiwan Center for Disease Control surveillance system. Counts were reported by age

group, sex, month, and county. Ages were reported as (0–4, 5–9, . . ., 65–69, 70+) years. The demographic data were obtained from the Department of Statistics, the Ministry of the Interior, Taiwan. These data included the mid-year population estimators, deaths, migration in single-year ages, and fertility in five-year age groups (15–19, . . ., 45–49). We used data in 2005–2018 as a training set. The demographic data from 2005 to 2017 were collected for the population demographic modelling (a shorter period because of the release schedule). All the training data in this article were published by the Taiwan officials and free access on the internet; the usage is licensed by the Open Government Data License: [https://data.gov.tw/license].

Importantly, we assumed no case detection gaps existed during the time frame covered by this article. We, therefore, regard "TB notification" and "TB incidence" as synonymous with the number culture-confirmed tuberculosis cases notified during a specific period.

## Age-specific incidence modelling and forecasting

We considered annual incidence rates by age and sex. The incidence rates by age groups and sex were calculated as the yearly notification counts divided by corresponding mid-year population estimates. Females and males were analysed separately with the same parameterisation. We modelled the incidence rates using Lee-Carter Models (LCMs) [19] formulated with age and time-varying terms. The LCMs were initially designed for mortality rate modelling, where they now predominate. Estimation, forecasting, bootstrapping methods for LCMs are well-developed.

We performed a likelihood-based LCM estimation, and also the comparable Poisson regression [20]:

$$\log(E(incident\ cases_{age,year})) = \alpha_{age} + \beta_{age}\kappa_{year} + \log(population_{age,year})$$

, where $E(.)$ is expectation function, $year \in \{2005, . . ., 2018\}$ is the calendar year, $\alpha_{age}$ is age effect term, $\kappa_{year}$ is period effect term, and $\beta_{age}$ is coefficients adjusting period effects for different age groups, and $age \in \{0-4, 5-9, . . ., 70+\}$ represents the age categories. To maintain identifiability, we imposed the constraints $\Sigma_{year}\kappa_{year} = 0$ and $\Sigma_{age}\beta_{age} = 1$. We followed the fitting procedure provided by Brouhns et al. [20] for likelihood maximisation.

Two nested Poisson models, one using an age-profile and a discrete period effect, i.e $\alpha_{age} + \kappa_{year}$, and another using an age-profile and a linear effect, i.e. $\alpha_{age} + year \times \kappa$, were used as comparators. Akaike information criterion (AIC), Bayesian information criterion (BIC), and log-likelihood were considered as goodness of fit metrics. The definitions of these metrics were identical to the ordinary Poisson regression model [21]. See S1 Appendix for the details of the LCM specification in our approach.

For forecasting, inspired by the Lee-Carter demographic forecasting, we used Autoregressive Integrated Moving Average models with drift [22], constructed from the LCM period effects. In forecasting, the death and birth processes applied semiparametric bootstrap sampling [23].

## Population modelling and forecasting

We constructed a synthetic population with birth, death and migration processes. The demographic methods adapted from those used in the Taiwan National Development Council's population projection report [7]. The demography was modelled by single age (0–100 years old) and sex. Mortality forecasting used the Lee-Carter model [19] below 84 years of age; and we used the Coale-Kisker method [24] for those aged over 85 years in death rate modelling as it was found to have a better reliability for small sample sizes in inferring of death rates. The birth forecasting used the fertility rates of women in childbearing ages, from 15 to 49, with a modified LCM [25]. For consistency with incidence forecasting, semiparametric bootstrap sampling was used for deaths and births [23]. The Migration process was modelled by linear regression with age effects and a

linear trend; the forecasting applied residual bootstrap sampling with the age-specific parameters seen in 2017. The forecasts were used for the next step by aggregating to the age groups as that of the incidence data. See S2 Appendix for the detailed methodology of the synthetic population.

### Forecasting overall TB incidence

The TB incidence model and the demographic model were built independently. Forecasts of age-specific TB incidence were weighted by forecasted population demography to obtain forecasts of per capita TB incidence for the whole population. TB incidence was calculated as per 100,000 rates by given strata. TB incidence rate reductions were calculated with respect to the incidence in 2015 and presented as percentages. For simplicity, some results were presented with age groups of 0–14, 15–34, 35–64, and above 65. In forecasting, the 95% prediction intervals and mean values were computed from 10,000 bootstrap samples. Uncertainty was propagated from every submodel. To compare with the global reduction target of the End TB strategy [2], we forecasted the incidence until 2035. The milestones of 2020, 2030 and 2035 of the End TB strategy of percentage reductions in per capita TB incidence from 2015 were used as intermediate outcomes.

### Incidence attributable to demographic change

We performed a scenario analysis to clarify the potential impact of demographic change. While forecasting the age-specific TB incidence to 2035, we kept the population size and age structure fixed as it was in 2018. This TB incidence was compared against values including projected changes in population structure by computing the fraction of total TB incidence attributable to demographic change in each year as $(I_{1,year} - I_{0,year})/I_{1,year}$, where $I_{1,year}$ and $I_{0,year}$ are the incident cases with and without demographic change respectively and is the calendar year. This corresponds to the definition of population attributable fraction [26].

All the analyses were performed using R 3.5 [27] and analysed/visualised by R package StMoMo, TSA, ggplot2 [28–30]. All analysis code is available at [https://github.com/TimeWz667/AgeingTB].

## Results

### Incidence modelling

Fig 1 shows the estimators of the Lee-Carter models of the incidence data. The age effect estimators ($\alpha_{age}$) suggested the baseline incidence rates increase with age. In both sexes, the higher levels in age groups older than fifteen years correspond to higher TB incidence rates. The point estimators of age-period adjustments ($\beta_{age}$) showed no specific trend. However, there are large uncertainties for all estimates pertaining to under 15-year age groups due to the small numbers of notifications observed. The period effect estimators ($\kappa_{year}$) had nearly constant slopes with calendar years. Fig 1 also demonstrates the forecasting of period effects with 95% prediction intervals: prediction intervals of both sexes grew at a constant rate with calendar time. Table 1 shows the goodness of fit of the LCMs, the nested age-period Poisson models, and age-trend Poisson models. In AIC, BIC and log-likelihood on the training data, the LCM result is preferred over the other two although it costs a higher degree of freedom. See S3 Appendix for the details of the goodness of fit, and residuals plots.

### Population forecasting

Fig 2 shows the demographic change from 2005 to 2035. In Fig 2A, the population will reach a maximum of 23.6 million in 2023, and will start shrinking to 23.2 million in 2035. The proportion of the population aged over 65 is increasing across the period and will reach 27% in 2035.

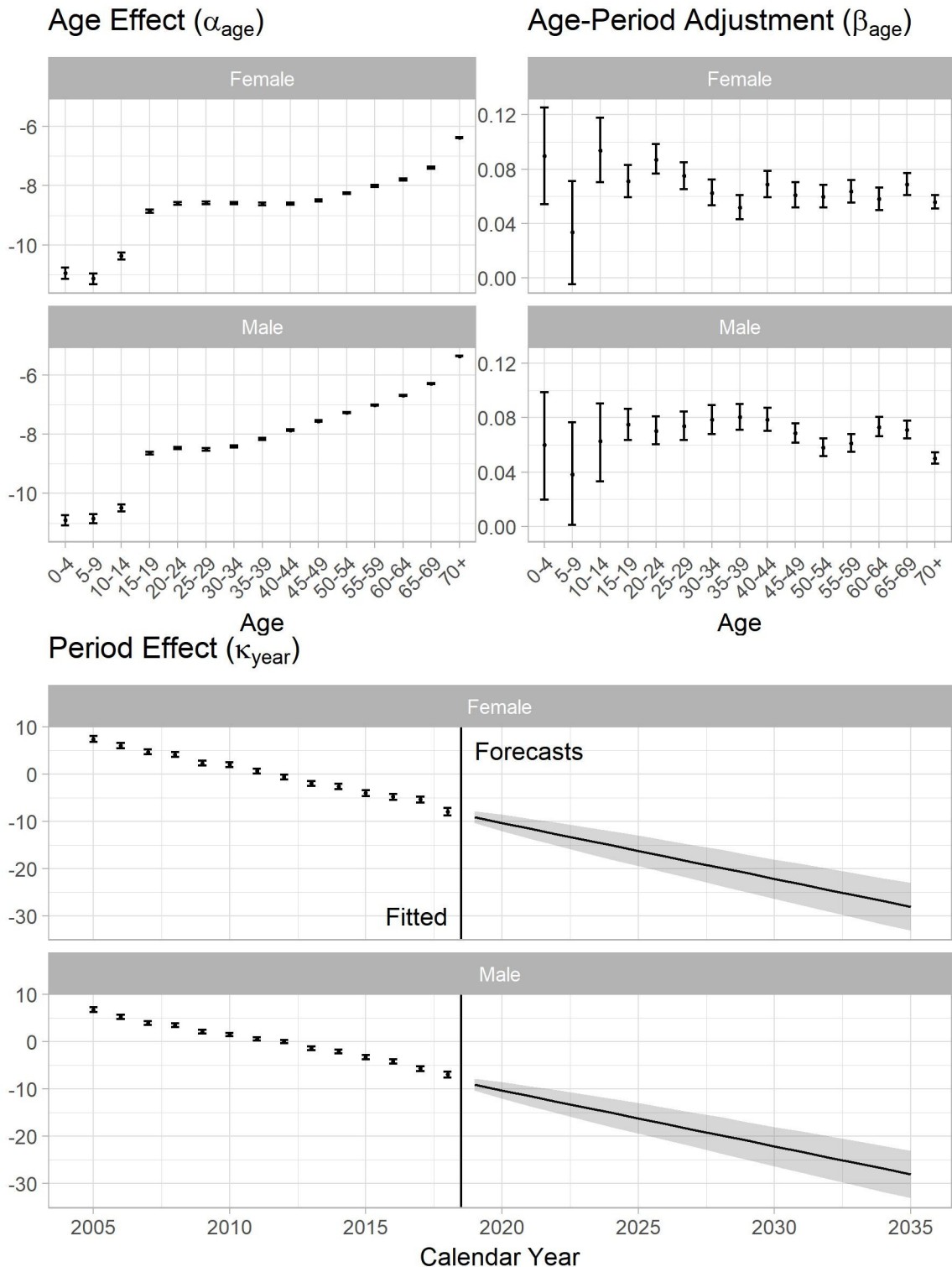

**Fig 1. Lee-Carter model fitting and forecasting of the TB incidence.** (Data: 2005–2018, Forecasting: 2019–2035). 95% confidence intervals of estimators and prediction intervals of forecasts were calculated through bootstrapping with 10,000 sample size.

**Table 1. Summary of model comparison.**

|  | Age-Trend | Age-Period | Lee-Carter Model |
|---|---|---|---|
| Model family | Poisson Regression | | |
| Period effect | Linear | Discrete | |
| No. observations | 420 | | |
| No. parameters | 32 | 56 | 84 |
| Log(Likelihood) | -1855 | -1819 | -1682 |
| AIC | 3773 | 3751 | 3531 |
| BIC | 3902 | 3977 | 3871 |

AIC: Akaike information criterion, BIC: Bayesian information criterion

The proportion of the population aged under 15 is declining to around 11%. Fig 2B compares the age structure of the Taiwanese population in 2018 and 2035, highlighting the population ageing.

### Incidence forecasting and age structure

Fig 3 demonstrates the trends of the population TB incidence rate and TB incidence rates by age-group (<15, 15–34, 35–64, >65). The forecast in Fig 3A suggests that the TB incidence in 2035 will be 22 (95% Prediction Interval (PI): 19–25) per 100,000. The overall incidence reduction will reach 54% (95% PI: 45%-59%) in 2035, which is 37% short of the reduction in the End TB Strategy. Fig 3B shows the age-specific incidence rates will have 60% to 80% reductions from 2015 to 2035 apart from the 5–9 group. The rate reductions in most age groups will be higher than the forecast reduction of 44% in the whole population. Fig 3C shows the overall incidence rates by age group as a stacked histogram. The TB incidence rates from age groups below 65 will be gradually decreasing whereas the above 65 will nearly stay constant from 2018 to 2035. Fig 3D shows the proportion of TB incidence in each age group. The proportion among adults aged over 65 years will reach 68% (95% PI: 67%-69%) and 79% (95% PI: 78%-81%) in 2025 and 2035, respectively. In 2035, more than 97% of incident cases will occur among those aged 35 years or older, indeed the contribution from cases under 15 years is nearly invisible in Fig 3C and 3D.

### Impact of demography on TB incidence

Fig 4 shows the forecast incidence rates with and without demographic change. In the scenario without demographic change, the forecast suggests that the incidence in 2035 will be around 13 per 100,000 compared to 23 with demographic change and the 90% reduction target of 4.5 per 100,000. The 95% prediction intervals for forecasts with demographic change continuously expand year by year, whereas without demographic change they converge to a constant width within five years. Table 2 shows the impact of demographic change. Up to 2020, TB incidence rates will have 23% and 27% reductions with and without demographic change respectively. Considering demographic change, the incidence rates are projected to reduce by 54% (95% PI: 45%-59%) from 2015 to 2035; without demographic change, the reduction will be 72% (95% PI: 67%-76%). In both scenarios, the trends of incidence rates showed diminishing reductions to the time scale. In both scenarios, the declines in incidence rates slowed over time. Comparing the forecasts with dynamic and fixed demography suggested that 39% (95% PI: 36%-42%) of incident TB cases in 2035 will be attributable to demographic change.

### Discussion

A substantial proportion of tuberculosis (TB) incidence in Taiwan is among people aged over 65 years. Social and economic development typically bring reductions in TB incidence but also

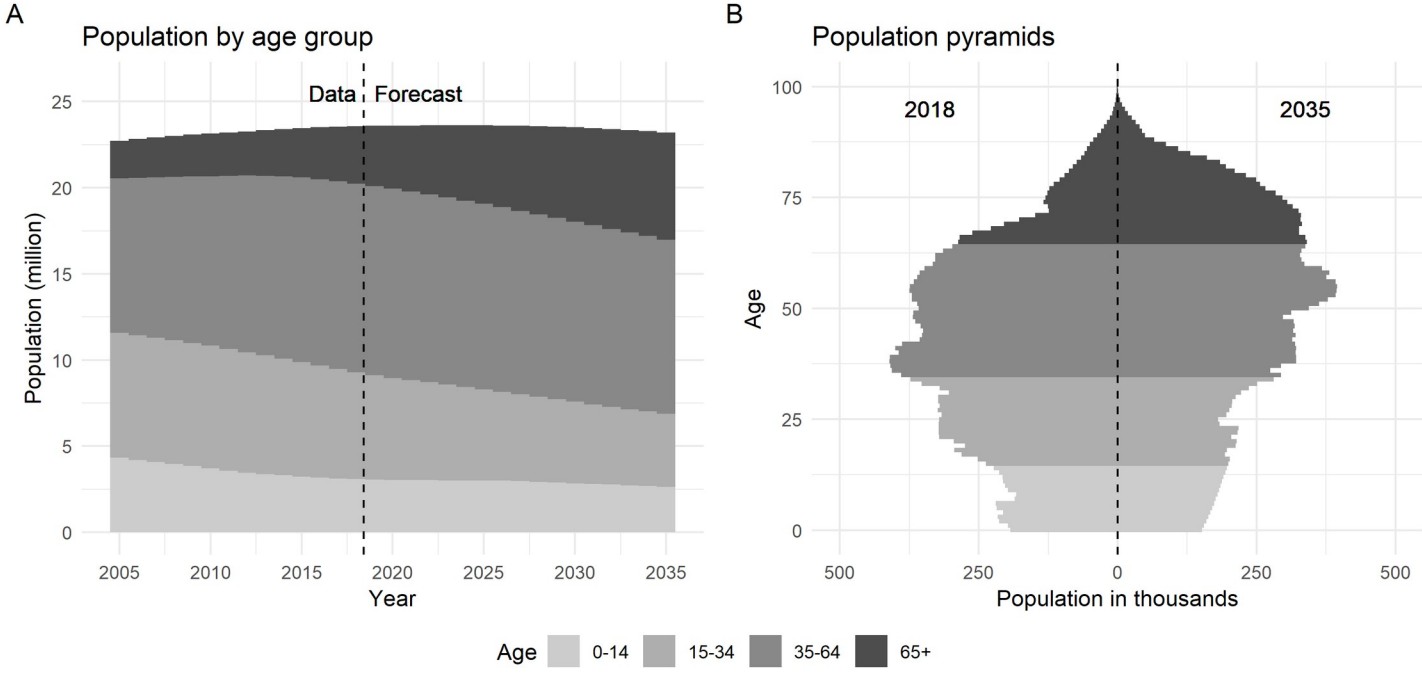

**Fig 2. Demographic change.** (Data: 2005–2017, Forecasting: 2018–2035).

reduced birth and death rates and population ageing. This study provides a novel investigation into the potential impact on TB incidence from population ageing using statistical modelling and forecasting. Current trends of TB incidence decline and demographic change suggest TB incidence rates in Taiwan will decrease to 25 per 100,000 by 2035. This represents a 45% reduction since 2015, missing the End TB goal of 90% reductions in TB incidence rates. We have shown that higher age-specific incidence rates in older age groups can mean that population ageing acts against reductions in TB rates, with TB incidence in 2035 projected to be 39% higher than without demographic change.

Previous studies have employed statistical methods either to forecast TB incidence, [12,13,31,32] or to analyse patterns by age using age-period-cohort models, [14,15] but we are the first study to statistically forecast age-specific TB incidence. Some transmission modelling studies [16,18] have explored issues related to age-structure, and Arregui et al. [18] generated forecasts. However, the fitting in Arregui et al. was not likelihood-based and did not use age-specific TB data, and so could not evaluate age-specific goodness of fit for TB projections or compare alternative models with conventional metrics. We made novel use of Lee-Carter models (LCMs), [19,22] which employ an elegant low-dimensional decomposition of age-specific rates to model trends and overall shape. LCMs were originally introduced for mortality rate modelling and are now the dominant approach, but have been applied elsewhere. Within demography, Hyndman [25] and Rueda-Sabater and Alvarez-Esteban [33] used LCMs to forecast the fertility rates, and Cowen [34] fitted LCMs to abortion rates. Kainz et al. [35] modelled chronic kidney disease prevalence as rate data, and Yue et al. [36] modelled cancer incidence and mortality. However, we are the first to apply LCMs to TB, finding they fitted better than Poisson Age-Period models. Our approach offers a generalizable and easily-implemented method for forecasting age-specific TB incidence and the impact of demographic change on total TB incidence.

In our model fitting results, the age effects ($\alpha_{age}$) demonstrated the TB incidence rates positively correlated with age in both females and males. The period effect estimators ($\kappa_{year}$) were

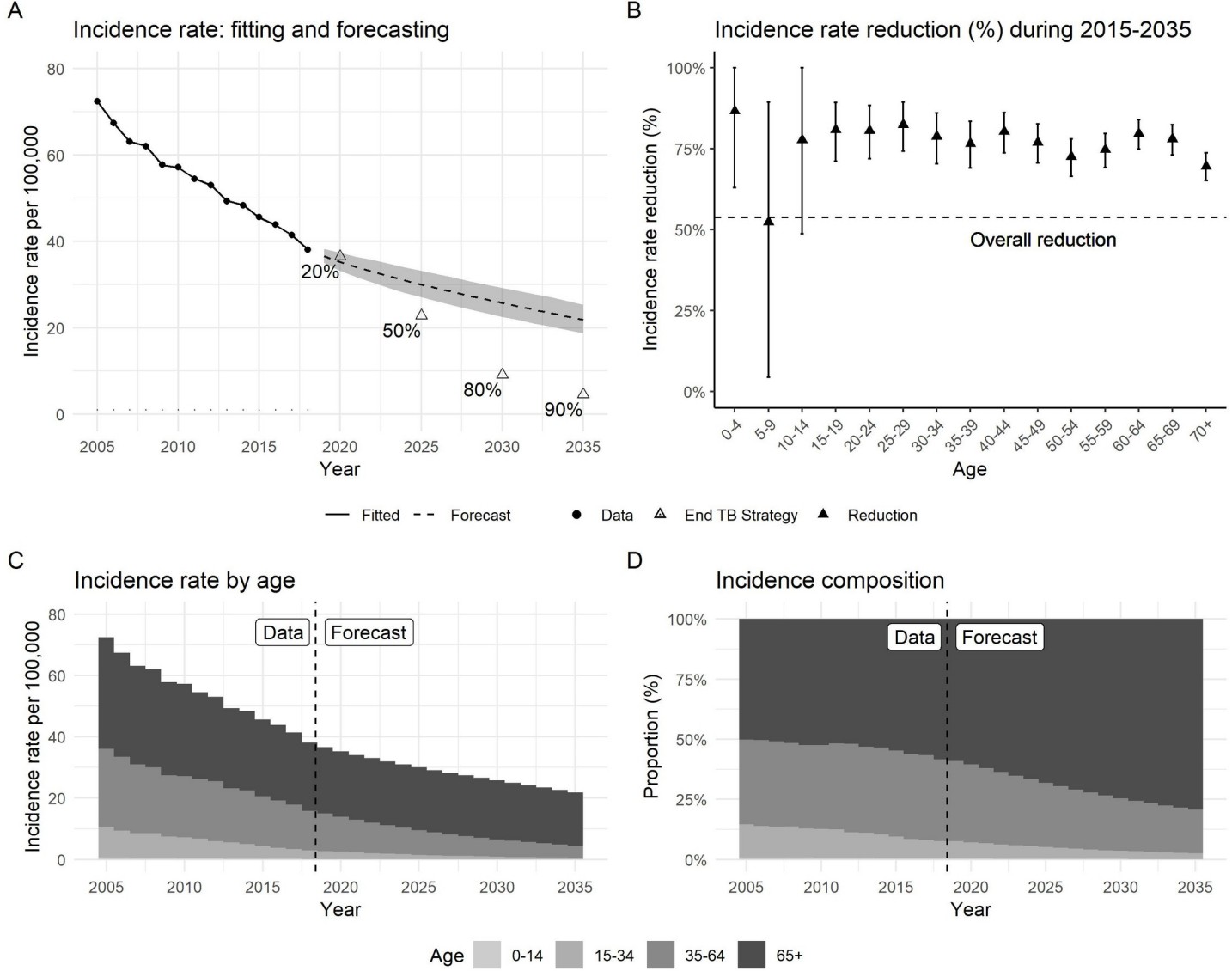

**Fig 3. TB incidence rate forecasting.** (A) Overall incidence rate per 100,000. In the forecasting, dashed line features the mean values and the shaded area is 95% prediction interval. (B) Incidence rate reductions by five-year age groups during 2015–2035 with 95% prediction interval. (C) Incidence rates attributed to age groups. (D) Proportions of age groups in Incidence cases.

almost linear despite not assuming linearity in the LCM formulation. The declines may reflect improvements in infection control and case detection, and the declining latent TB prevalence in each age group. Improvements in infection control and case detection both reduce the force of infection that will induce further TB incidence. Latent TB prevalence depends on cumulative lifetime infection risks, and therefore on the history of active TB prevalence. As TB incidence has been declining, the latent TB prevalence in recent cohorts will be lower than in historical cohorts at the same age. Lastly, the age-period interaction terms ($\beta_{age}$) were used to demonstrate how the incidence rate reduced differently in each age group, although no overall pattern was identified. The variance of the estimators in young people was larger because the only around 1% of incident TB (< 100 cases every year in the recent decade) were from people below 15 years old.

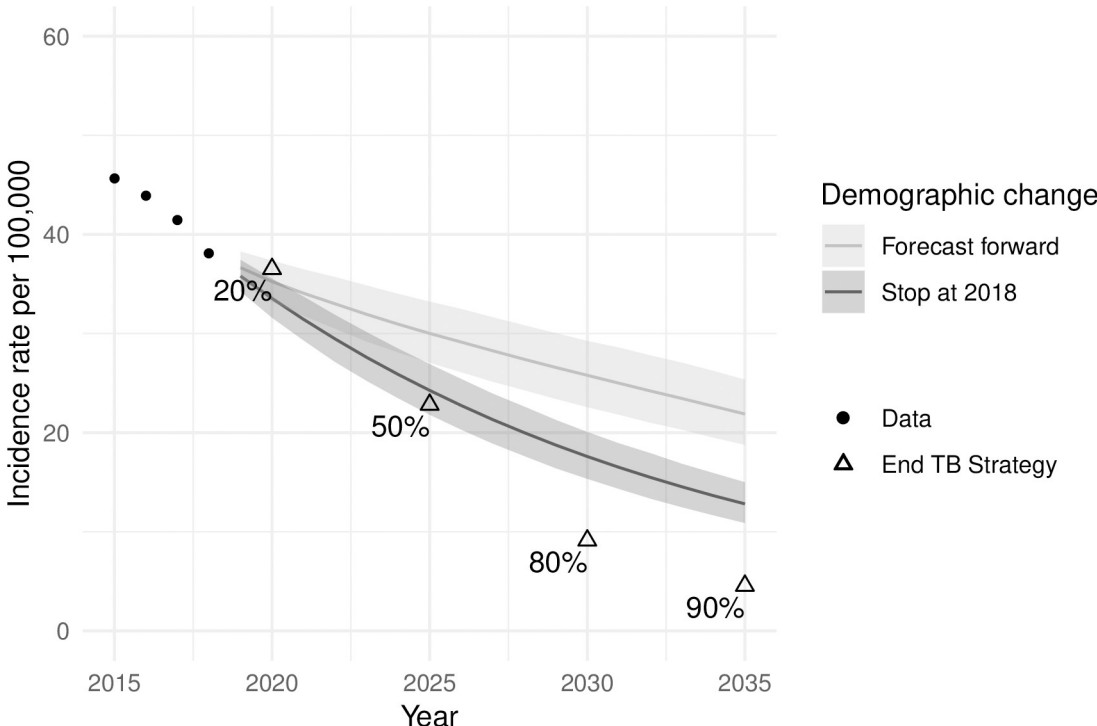

**Fig 4. TB incidence with and without demographic change.** Ribbons show 95% prediction intervals.

For Taiwan and many other high-income settings, TB notifications are thought to parallel TB incidence with only a small gap. Taiwan lacks survey data (e.g. capture-recapture studies) to directly inform on the magnitude of this gap. In settings where this gap is larger and changing over time, interpretation of TB notification data is more problematic, and notifications may not be a good proxy for incidence. Even in a declining TB epidemic with evolving case-mix, it is possible that case detection may change differently over time in different age-groups; we have not attempted to include such effects. Taiwan does not have United Nations Population Division demographic forecasts, hence our bespoke demographic modelling. For most nations, these forecasts could be used 'off the shelf'. We have presented results on percentage reductions in both per capita TB incidence rates and in total TB incidence (e.g. Table 2), which are similar because of Taiwan's small projected population change over the period considered; this may not be true in all settings.

The decline in TB incidence in Taiwan probably has multiple contributory causes, including improvements in TB control, socio-economic development, and the reductions in the prevalence of latent TB as a result of declining transmission. For an infectious disease like TB, reduced transmission can amplify and sustain over time changes in underlying causative factors, complicating their analysis. The low TB rates in children aged under fifteen may reflect

**Table 2. Summary of reductions in TB incidence with and without demographic change.**

| Year | Percentage reduction in per capita TB incidence from 2015: | | Percentage of total TB incidence attributable to demographic change |
|------|------------------------|------------------------------------------|-------------------------------------------------|
|      | with demographic change | without demographic change after 2018 |  |
| 2020 | 23.1% (18.2%, 27.2%) | 26.6% (22.2%, 30.8%) | 4.6% (1.6%, 7.5%) |
| 2025 | 35.2% (27.2%, 40.7%) | 47.0% (41.0%, 52.2%) | 18.3% (15.5%, 21.2%) |
| 2030 | 45.0% (35.9%, 50.5%) | 61.6% (56.0%, 66.4%) | 30.0% (27.3%, 33.0%) |
| 2035 | 53.7% (44.5%, 58.9%) | 72.1% (67.1%, 76.1%) | 38.8% (36.1%, 41.7%) |

low exposure to TB in this group or potentially lower rates of case detection. Our assessment of the impact of population ageing on TB incidence and case-mix has particular current relevance to many WHO Western Pacific region countries [1] and will be relevant to many more countries and regions in the future. Our analysis could provide a template for analysts who wish to explore issues related to future TB incidence and demography where age-specific data are available.

Our analysis accounted for cohort propagation of latent tuberculosis infection (LTBI) in a phenomenological way. LTBI represents accumulated lifetime risk of infection by exposure to active tuberculosis disease. Older individuals in most settings have higher LTBI prevalence due both to longer cumulative exposure and (in declining epidemics) exposure to a higher mean infection rate over their lifetime. The ageing through of these LTBI positive cohorts thus generates a secular time trend in reactivation disease rates at a particular age. Our approach does not explicitly model LTBI prevalence, because this would introduce additional parameters and, without LTBI data, identifiability issues. However, LTBI cohort effects are accounted for in our current approach indirectly by modelling the secular trends in age-specific incidence rates.

Another limitation of our approach is that it would fail to account for non-linear threshold behaviour, such as during outbreaks. However, in many high-income settings (including Taiwan), the steadily declining tuberculosis incidence implies the net reproduction number is below one.

Extending the model by adding exogenous variables is possible. Our analysis did not address the impact of other variables for simplicity and clarity. Important predictors could include socioeconomic status and comorbidities such as diabetes mellitus [37]. However, projections would require additional time-series analysis to forecast these explanatory variables. It is worth noting that according to Taiwan CDC surveillance, in 2005, 0.72% of TB cases in Taiwan were coded as HIV; neglecting HIV is unlikely to have impacted our results.

Older age as a risk factor for TB disease has perhaps been under-explored since age is not a modifiable risk factor, and since in most current high-burden settings populations and the typical age of TB cases are fairly young. Our result that population ageing will act to slow declines in TB incidence tallies with that of Arregui et al [18], obtained for different settings using very different methods, and quantifies the magnitude of this effect. However, the importance of older age groups to TB control is already evident in many Asian populations, [16] and this will be an increasingly widespread facet of global TB control if reductions in incidence continue and accelerate in the future. Older populations will also have their own particular challenges in terms of access, diagnosis and comorbidities complicating their care. Public health planning to develop adapted strategies for care and control to meet these changing population needs is essential.

In conclusion, the Lee-Carter model provides a tool to project age-specific tuberculosis incidence and hence forecast overall TB incidence while accounting for demographic change. In Taiwan, population ageing may slow the decline of TB incidence by 39% over the period 2015–2035. TB care and control programmes will increasingly need to address the needs of older adults, who will comprise a growing majority of the TB epidemic.

## Supporting information

**S1 Appendix. Specification of the Lee-Carter approach.**
(PDF)

**S2 Appendix. Synthetic population, modelling details.**
(PDF)

**S3 Appendix. Residual analysis and the goodness of fit.**
(PDF)

## Author Contributions

**Conceptualization:** Chu-Chang Ku.

**Data curation:** Chu-Chang Ku.

**Formal analysis:** Chu-Chang Ku.

**Funding acquisition:** Peter J. Dodd.

**Investigation:** Chu-Chang Ku.

**Methodology:** Chu-Chang Ku, Peter J. Dodd.

**Project administration:** Chu-Chang Ku.

**Resources:** Chu-Chang Ku.

**Software:** Chu-Chang Ku.

**Supervision:** Peter J. Dodd.

**Validation:** Peter J. Dodd.

**Visualization:** Chu-Chang Ku.

**Writing – original draft:** Chu-Chang Ku, Peter J. Dodd.

**Writing – review & editing:** Chu-Chang Ku, Peter J. Dodd.

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
