## [Decision Letter · Decision Letter 0]

24 Jul 2019

PONE-D-19-18299

Forecasting the impact of population ageing on tuberculosis incidence

PLOS ONE

Dear Mr. Ku,

Thank you for submitting your manuscript to PLOS ONE. After careful consideration, we feel that it has merit but does not fully meet PLOS ONE’s publication criteria as it currently stands. Therefore, we invite you to submit a revised version of the manuscript that addresses the points raised during the review process.

We would appreciate receiving your revised manuscript by Sep 07 2019 11:59PM. To enhance the reproducibility of your results, we recommend that if applicable you deposit your laboratory protocols in protocols.io, where a protocol can be assigned its own identifier (DOI) such that it can be cited independently in the future. For instructions see: http://journals.plos.org/plosone/s/submission-guidelines#loc-laboratory-protocols

We look forward to receiving your revised manuscript.

Kind regards,

Michele Tizzoni

Academic Editor

PLOS ONE

Journal Requirements:

2) Please add the following to your COI statement: "Peter J. Dodd is also a PLOS ONE handling editor for this collection."

Please confirm that this does not alter your adherence to all PLOS ONE policies on sharing data and materials, by including the following statement: "This does not alter our adherence to  PLOS ONE policies on sharing data and materials.” (as detailed online in our guide for authors http://journals.plos.org/plosone/s/competing-interests).  

If there are restrictions on sharing of data and/or materials, please state these. Please note that we cannot proceed with consideration of your article until this information has been declared.

Reviewers' comments:

Reviewer's Responses to Questions

**Comments to the Author**

1. Is the manuscript technically sound, and do the data support the conclusions?

Reviewer #1: Yes

Reviewer #2: Yes

2. Has the statistical analysis been performed appropriately and rigorously? 

Reviewer #1: Yes

Reviewer #2: Yes

3. Have the authors made all data underlying the findings in their manuscript fully available?

Reviewer #1: Yes

Reviewer #2: Yes

4. Is the manuscript presented in an intelligible fashion and written in standard English?

Reviewer #1: Yes

Reviewer #2: Yes

5. Review Comments to the Author

Reviewer #1: The authors estimate future trends in the age-specific tuberculosis (TB) incidence in Taiwan until 2035, by applying statistical models (called Lee-Carter models) to time series of incidence data from 2007-2017 and accounting for sex, age, and year-specific effects. The estimated age-specific incidences are then applied to the projected demographic age structure to derive total incidence trends. Overall, the idea is simple and the considerations on demographic changes are important for projections of TB incidence in countries with declining transmission and undergoing a demographic transition, therefore the paper deserves consideration.

My main criticism is that Lee-Carter models were, according to the authors, previously applied in demography and non-communicable diseases epidemiology. However, because the incidence of infectious diseases is critically dependent on the current prevalence of infections, highly non-linear effects in temporal trends take an important role (e.g. threshold effects in the reproductive number) which may make these models highly inaccurate for projecting disease incidence. If transmission dynamics are very far from the epidemic threshold and therefore the large majority of TB cases are due to reactivations, these effects can likely be neglected. Previous studies from US (a low incidence country) showed that only 30% of cases are recently transmitted (Guzzetta et al., JTB 2011), and the large burden of reactivated disease results in a poor performance of TB control strategies applied in the state (Guzzetta et al., JTB 2013). Considering that the average incidence of TB in the US is around 5 per 100k, one might expect a larger contribution of recently transmitted cases in Taiwan, given the average current incidence of about 8 times the US one.

Thus, the authors should:

- provide a demonstration that the method works when parametrizing the model with a subset of data, e.g. the first 5 years of the time series, and projecting the final 5 years against actual observations (this would not be final proof, because short term projections are highly correlated to the current status, but would at least show that the model works in a very easy case);

- discuss the proportion of reactivated vs. recently transmitted TB in Taiwan or countries with similar epidemiology/geographic settings, in light of the above criticism;

- acknowledge the overall limitation of using Lee-Carter models in a context of highly non-linear time trends.

Other comments:

- a minimum description of how Lee-Carter models work mathematically should be provided, at least in the Supplementary Materials; a Coale-Kisker method is mentioned at l. 147, but no specification of why a different method is used for the age class above 85 years old, nor a description of how the method differs from the Lee-Carter one;

- the definition of the likelihood formula used to calculate AIC and BIC should be made explicit;

- do the authors have specific reasons for considering sex-specific incidence separately? Are differences in results significant? Fig. 1 could show estimates for the two sexes in the same graph, in order to allow an easier comparability;

- l. 59: the burden of TB in the mentioned countries cannot be considered "high" from a global point of view. They are perhaps higher than most industrialized countries, but certainly very low compared to the 22 high-burden countries representing 83% of the overall TB burden. Also, "Korean" should be South Korea;

- l. 189: "excepting the reference group aged 0-4": what is the exception? Uncertainties seem large for this age group as well;

- l. 190: "constant trends with calendar years": the trend is linear, perhaps the authors mean that the effect (slope) is constant;

- l. 219, "which is 37% short of the 90%": using percentages as differences may be very confusing, I recommend dropping the 37% figure;

- sentence at lines 246-247 is not clear;

- when mentioning co-morbidities in the discussion (l.328 and 329), HIV should also be mentioned and average prevalences/trends in Taiwan of the two main comorbidities should be provided, to give a general idea on how much neglecting them can impact results;

- I suggest to move the final paragraph of the discussion to somewhere in the beginning: as it is, it has a very anticlimactic effect.

Reviewer #2: This work presents a series of analyses based on statistical modelling to describe the interplay between population's ageing and TB incidence rates -both aggregated and age-specific- in Taiwan. Based on Lee-Carter models, authors analyse age-specific time series between 2005 and 2018 regarding both TB incidence and population structures, and extrapolate to produce forecasts of both aggregated and age-specific TB incidence rates that run until 2035.

The paper is written in a clear and concise manner, and the general research question -what can be expected from the effects of populations' aging on global TB burden levels- is timely and of utmost importance. Statistical modelling methods are sound and described in a (perhaps too much) succinct way. Results -the main observation that contemplating populations' ageing translate into more pessimistic forecasts for TB incidence in Taiwan for the next years- are robust, and in line with previous literature, which is, however, scarce, as authors point out. Conclusions are backed up by the analyses done, and the limitations of the statistical approach are framed in a way that is essentially adequate.

I have therefore no major objections for the publication of this manuscript, and congratulate the authors for their important work. I have, though, the following minor comments/questions, which should be successfully addressed before I can finally recommend the article for publication in this journal:

1. I think that the description of the methods should probably be more exhaustive and explicit, given the specialised character of the statistical modelling framework used in this study, which the interdisciplinary audience of PLoS One might not be necessarily familiar with. Other aspects that might better be explained to a higher level of detail are how the age-specific and aggregated TB incidence rates are built, and rescaled from the demographic and migration forecasts, how does the bootstrap work and how (explicitly) does the uncertainty to TB rates propagates from the different sub-models.

2. At several points of the manuscript, we read the following statements:

line 74: "time series analysis producing age-specific forecasts of the TB incidence has not been published to our knowledge."

line 83: "However, age-specific forecasting and the impact of demographic change have yet to be analysed."

line 281: "Some transmission modelling studies [16,18] have explored issues related to age-structure, but without forecasts or formal assessment of fit."

line 335: "Our result that population ageing will act to slow declines in TB incidence does not seem to have been previously noted."

Which are not totally true. As a matter of fact, reference [18] is a study where authors report the impact of populations' ageing on TB incidence forecasts using transmission modelling. In [18], incidence rate forecasts, both aggregated and age-specific, are indeed reported for different countries, as well as fit evaluations of incidence and mortality rates between 2000 and 2015, upon model calibration. Importantly, the main conclusion of that work -that populations' ageing appears to be directly proportional to an increase in model-based TB burden forecasts with respect to simpler estimations that neglect demographic evolution- is exactly the same of the work here presented for the case of Taiwan, despite the type of models used in that work being radically different from what is presented here. Therefore, the aforementioned statements should be modified, and the findings presented in this work should be put in context to the conceptually similar results reported for other countries in [18].

3. In their analyses, authors assume, as they explicitly acknowledge, the equivalence between TB cases notifications and incidence for the sake of the results they reports. Is not there available data about case notification rates that could be integrated into the models? If not, this possibility should at least be discussed. Even if working with just notification data might be reasonable in the case of contemporary Taiwan; the changes in the population structure that authors forecast in the years to come, along with the eventual added difficulties to detect and register active TB cases in the oldest population strata (which authors also discuss in the introduction) might translate into the growth of a reservoir of undetected/unregistered active TB among eldest age-groups. This plausible scenario might bias the quantitative conclusions of this work, and it should probably be discussed when exposing the limitations of working on notification data alone.

-3. In line 224, we read: "age groups below 65 will be gradually decreasing whereas the above 65 will nearly stay constant from 2018 to 2035"

It took me some time to understand that the age-specific incidence is proportional to the area under the curves, but not to the lines (i.e. that the histograms are stacked), this probably should be stated more clearly. Also, and more important, in figures 3C-3D, four shades are included in the legend, but only three can be appreciated in the figures.

4. The text is very well written, I only found the following couple of typos:

Line 195: "although it cost a higher degree of freedom" should read "it costs"

Line 262: "Table 2. Summary of reductions in TB incidence reductions with and without

263 demographic change" (remove the second "reductions"?)

6. PLOS authors have the option to publish the peer review history of their article (what does this mean?). If published, this will include your full peer review and any attached files.

Reviewer #1: No

Reviewer #2: No

---

## [Author Response · Author response to Decision Letter 0]

19 Aug 2019

Reviewer #1: The authors estimate future trends in the age-specific tuberculosis (TB) incidence in Taiwan until 2035, by applying statistical models (called Lee-Carter models) to time series of incidence data from 2007-2017 and accounting for sex, age, and year-specific effects. The estimated age-specific incidences are then applied to the projected demographic age structure to derive total incidence trends. Overall, the idea is simple and the considerations on demographic changes are important for projections of TB incidence in countries with declining transmission and undergoing a demographic transition, therefore the paper deserves consideration.

My main criticism is that Lee-Carter models were, according to the authors, previously applied in demography and non-communicable diseases epidemiology. However, because the incidence of infectious diseases is critically dependent on the current prevalence of infections, highly non-linear effects in temporal trends take an important role (e.g. threshold effects in the reproductive number) which may make these models highly inaccurate for projecting disease incidence. If transmission dynamics are very far from the epidemic threshold and therefore the large majority of TB cases are due to reactivations, these effects can likely be neglected. Previous studies from US (a low incidence country) showed that only 30% of cases are recently transmitted (Guzzetta et al., JTB 2011), and the large burden of reactivated disease results in a poor performance of TB control strategies applied in the state (Guzzetta et al., JTB 2013). Considering that the average incidence of TB in the US is around 5 per 100k, one might expect a larger contribution of recently transmitted cases in Taiwan, given the average current incidence of about 8 times the US one.

Thanks for this point. It is indeed a limitation that this projection method would not take into account threshold effects. We agree that the proportion of incidence due to reactivation is a valuable metric, but note that it is not logically connected with threshold behaviour (eg a high prevalence setting with declining incidence could have a majority of incidence from recent transmission but will, by definition, have a net reproduction number below one). We have added the following text to the limitations Discussion:

“Another limitation of our approach is that it would fail to account for non-linear threshold behaviour, such as during outbreaks. However, in many high-income settings (including Taiwan), the steadily declining tuberculosis incidence implies the net reproduction number is below one.”

Also we have added a paragraph in the discussion to emphasise the explanation of our model fitting results.

“In our model fitting results, the age effects demonstrated the TB incidence rates positively correlated with age in both females and males. The period effect estimators were almost linear despite not assuming linearity in the LCM formulation. The declines may reflect improvements in infection control and case detection, and the declining latent TB prevalence in each age group. Improvements in infection control and case detection both reduce the force of infection that will induce further TB incidence. For latent TB, which is accumulated during one's lifetime and depends on historical TB prevalent TB in history, different cohorts will have different prevalence. As TB incidence has been declining, the latent TB prevalence in recent cohorts will be lower than in historical cohorts at the same age. Lastly, the age-period interaction terms were used to demonstrate how the incidence rate reduced differently in each age group, although no overall pattern was identified. The variance of the estimators in young people was larger because the only around 1% of incident TB (< 100 cases every year in the recent decade) were from people below 15 years old.

” 

Thus, the authors should:

- provide a demonstration that the method works when parametrizing the model with a subset of data, e.g. the first 5 years of the time series, and projecting the final 5 years against actual observations (this would not be final proof, because short term projections are highly correlated to the current status, but would at least show that the model works in a very easy case);

We have added a validation figure in S3 appendix to demonstrate the goodness of fit 

- discuss the proportion of reactivated vs. recently transmitted TB in Taiwan or countries with similar epidemiology/geographic settings, in light of the above criticism;

Please see above. We have added a sentence pointing out the limitation that this method would not correctly predict threshold behaviour, and noting that for the settings where this method is most applicable, and the questions we investigate of most interest, incidence has been declining for a decade or more suggesting that the net reproduction number is safely below one.

We would expect our method to apply in high-burden settings with declining epidemics also, which might still have the majority of incidence due to recent transmission but still have R<1. But for these settings, the equivalence of notifications and incidence is not usually a safe assumption. 

For your interest, estimates from our unpublished tuberculosis transmission modelling in Taiwan suggest around 50% of cases are recently transmitted in 2018 and will be down to 30% in 2035 .

- acknowledge the overall limitation of using Lee-Carter models in a context of highly non-linear time trends.

We have added the limitation of not being able to predict non-linear threshold behaviour in the Discussion (see above) - thanks for pointing this out. However, we would like to emphasize that the LCM we use is not restricted to linear trends in time (if that is what was meant). We have clarified this as part of our more detailed description of the LCM approach by adding an appendix file, S1 Appendix, with details of the LCM specification.

Other comments:

- a minimum description of how Lee-Carter models work mathematically should be provided, at least in the Supplementary Materials; a Coale-Kisker method is mentioned at l. 147, but no specification of why a different method is used for the age class above 85 years old, nor a description of how the method differs from the Lee-Carter one;

Thanks for the suggestion to describe LCMs in more detail. Reviewer 2 also suggested this and we agree. We have added a supplementary, S2 Appendix, to list the implementation details of our synthetic population modelling. 

Thanks also for highlighting the unjustified detail around motivating the Coale-Kisker method in our demographic model. We have added:

“we used the Coale-Kisker method [24] for those aged over 85 years in death rate modelling as it was found to have a better reliability for small sample sizes in inferring of death rates. ”

- the definition of the likelihood formula used to calculate AIC and BIC should be made explicit;

Thanks. We have added the definitions as in S1 Appendix. 

- do the authors have specific reasons for considering sex-specific incidence separately? Are differences in results significant? Fig. 1 could show estimates for the two sexes in the same graph, in order to allow an easier comparability;

Thanks for the question. The motivation for modelling sex-specific incidence was primarily because in Taiwan, as in most settings, the majority of incidence is in men. The reasons for this are complex and incompletely understood (see eg Horton et al). We didn’t want to assume that the trends would be the same in each sex and so modelled them separately. 

- l. 59: the burden of TB in the mentioned countries cannot be considered "high" from a global point of view. They are perhaps higher than most industrialized countries, but certainly very low compared to the 22 high-burden countries representing 83% of the overall TB burden. Also, "Korean" should be South Korea;

Thanks, a very reasonable point. We have changed “high TB burden” to “TB burdens of substantial public significance”. We have changed “Korean” to “South Korea”. 

For information, WHO have discontinued use of their 22 high burden country (HBC) list; their new 30 HBC tuberculosis incidence list includes China due to its large absolute incidence. 

- l. 189: "excepting the reference group aged 0-4": what is the exception? Uncertainties seem large for this age group as well;

Thanks, the statement was a mistake. To fix this, we have removed the clause “excepting the reference group aged 0-4”; the sentence is still true. The uncertainties in children were due to the low number of notifications observed; we have added “due to small numbers.” in lieu of the previous clause.

- l. 190: "constant trends with calendar years": the trend is linear, perhaps the authors mean that the effect (slope) is constant;

Fixed

- l. 219, "which is 37% short of the 90%": using percentages as differences may be very confusing, I recommend dropping the 37% figure;

We take your point. We decided to drop the 90% so as to still include our numerical result here, ie:

“which is 37% short of the reduction in the End TB Strategy.”

- sentence at lines 246-247 is not clear;

Thanks for catching this. We have revised to:

“In both scenarios, the declines in incidence rates slowed over time. Comparing the forecasts with dynamic and fixed demography suggested that 39% (95% PI: 36%-42%) of incident TB cases in 2035 will be attributable to demographic change.”

- when mentioning co-morbidities in the discussion (l.328 and 329), HIV should also be mentioned and average prevalences/trends in Taiwan of the two main comorbidities should be provided, to give a general idea on how much neglecting them can impact results;

Thanks. We have added:

“It is worth noting that according to Taiwan CDC surveillance, in 2005, 0.72% of TB cases in Taiwan were coded as HIV; neglecting HIV is unlikely to have impacted our results.”

- I suggest to move the final paragraph of the discussion to somewhere in the beginning: as it is, it has a very anticlimactic effect.

Apologies. We were intending the last paragraph to be a ‘Conclusions’ as is commonly found in the biomedical literature, and may be sought for by readers with this background. To flag this more explicitly, we have changed this paragraph to begin “In conclusion,”, but left it where it was.

Reviewer #2: This work presents a series of analyses based on statistical modelling to describe the interplay between population's ageing and TB incidence rates -both aggregated and age-specific- in Taiwan. Based on Lee-Carter models, authors analyse age-specific time series between 2005 and 2018 regarding both TB incidence and population structures, and extrapolate to produce forecasts of both aggregated and age-specific TB incidence rates that run until 2035.

The paper is written in a clear and concise manner, and the general research question -what can be expected from the effects of populations' aging on global TB burden levels- is timely and of utmost importance. Statistical modelling methods are sound and described in a (perhaps too much) succinct way. Results -the main observation that contemplating populations' ageing translate into more pessimistic forecasts for TB incidence in Taiwan for the next years- are robust, and in line with previous literature, which is, however, scarce, as authors point out. Conclusions are backed up by the analyses done, and the limitations of the statistical approach are framed in a way that is essentially adequate.

I have therefore no major objections for the publication of this manuscript, and congratulate the authors for their important work.

Thank you for your kind comments.

 I have, though, the following minor comments/questions, which should be successfully addressed before I can finally recommend the article for publication in this journal:

1. I think that the description of the methods should probably be more exhaustive and explicit, given the specialised character of the statistical modelling framework used in this study, which the interdisciplinary audience of PLoS One might not be necessarily familiar with. Other aspects that might better be explained to a higher level of detail are how the age-specific and aggregated TB incidence rates are built, and rescaled from the demographic and migration forecasts, how does the bootstrap work and how (explicitly) does the uncertainty to TB rates propagates from the different sub-models.

Thanks for this comment. Reviewer 1 also requested additional detail in the methodology (especially around LCMs). In response to their comments and yours, we have made the following changes:

Additional detail around LCM specification in S1 Appendix, including modelling fitting procedure, likelihood function, deviance residuals, bootstrap procedure, and propagation of uncertainty. The implementation can also be found in the code base we mentioned. 

We hope these changes have improved the text without introducing too much detail for the typical reader. We note that additional detail is available in the supplementary appendix and that code for the analyses is fully open source and available as described.

2. At several points of the manuscript, we read the following statements:

line 74: "time series analysis producing age-specific forecasts of the TB incidence has not been published to our knowledge."

line 83: "However, age-specific forecasting and the impact of demographic change have yet to be analysed."

line 281: "Some transmission modelling studies [16,18] have explored issues related to age-structure, but without forecasts or formal assessment of fit."

line 335: "Our result that population ageing will act to slow declines in TB incidence does not seem to have been previously noted."

Which are not totally true. As a matter of fact, reference [18] is a study where authors report the impact of populations' ageing on TB incidence forecasts using transmission modelling. In [18], incidence rate forecasts, both aggregated and age-specific, are indeed reported for different countries, as well as fit evaluations of incidence and mortality rates between 2000 and 2015, upon model calibration. Importantly, the main conclusion of that work -that populations' ageing appears to be directly proportional to an increase in model-based TB burden forecasts with respect to simpler estimations that neglect demographic evolution- is exactly the same of the work here presented for the case of Taiwan, despite the type of models used in that work being radically different from what is presented here. Therefore, the aforementioned statements should be modified, and the findings presented in this work should be put in context to the conceptually similar results reported for other countries in [18].

Many apologies. On re-reading we do indeed do a disservice to reference 18 (this was a late addition in redrafting, as our systematic review of relevant literature focussed on statistical models). 

In response to this:-

We have changed the last sentence of the Introduction paragraph (previously ending with the sentence you note as line 83) to:

“Indeed, Arregui et al. [18] developed forecasts for the effects of demographic change on TB epidemics, focussing on four relatively young countries; our interests are in developing statistically rigorous time-series approaches and in focusing on an example of a much older population.”

(We have left the sentence starting this paragraph - you reference as line 74 - as it was, since we believe this is correct.)

We have also changed a sentence in the last paragraph of the Introduction (by adding more):

“...so more methods to understand and forecast the impact of changing demography on TB epidemics are needed.”

We have changed the sentence you refer to at line 281 to:

“...Some transmission modelling studies [16,18] have explored issues related to age-structure, and Arregui et al [18] generated forecasts. However, the fitting in Arregui et al was not likelihood-based and did not use age-specific TB data, and so could not evaluate age-specific goodness of fit for TB projections or compare alternative models with conventional metrics.”

We have changed the sentence you refer to at line 335 to:

“...Our result that population ageing will act to slow declines in TB incidence tallies with that of Arregui et al [18], obtained for different settings using very different methods, and quantifies the magnitude of this effect...”

3. In their analyses, authors assume, as they explicitly acknowledge, the equivalence between TB cases notifications and incidence for the sake of the results they reports. Is not there available data about case notification rates that could be integrated into the models? If not, this possibility should at least be discussed. Even if working with just notification data might be reasonable in the case of contemporary Taiwan; the changes in the population structure that authors forecast in the years to come, along with the eventual added difficulties to detect and register active TB cases in the oldest population strata (which authors also discuss in the introduction) might translate into the growth of a reservoir of undetected/unregistered active TB among eldest age-groups. This plausible scenario might bias the quantitative conclusions of this work, and it should probably be discussed when exposing the limitations of working on notification data alone.

Unfortunately, there is no data in Taiwan to directly inform on the gap between notifications and incidence. This is a general problem for tuberculosis. Measuring tuberculosis incidence in the general population is not feasible and has never been done at a nationally representative level. Nationally representative tuberculosis prevalence surveys provide an unbiased measure of disease burden (though typically with around a 20% error margin), but the relationship between tuberculosis prevalence and incidence is uncertain and not constant across settings. Inventory studies are useful, especially for quantifying the contribution of under-reporting (as opposed to under-diagnosis) to the gap between notifications and incidence, and where three independent records are available they can be specialized to capture-recapture studies that inform on the whole notification-incidence gap (under additional assumptions). Unfortunately, a tuberculosis prevalence survey in Taiwan is unlikely to be pursued (eg they fail to meet the burden threshold above which WHO suggest such a survey would be useful and would need a huge sample size), and the universal insurance system means that the conditions for a capture-recapture study are unlikely to be met.

In the absence of these types of survey data, the recourse is typically to expert opinion on the case-detection ratio or a standard assumption about the notification-indence gap. The above hierarchy of approaches is broadly the approach followed by WHO in estimating tuberculosis incidence for member states (of which Taiwan is not one). Your suggestion that changing demography and increasingly rare cases may undermine experience in detecting tuberculosis in some groups, reversing improvements in case-detection is an interesting one, which although it has been made elsewhere, lacks data to strongly support it.

Obviously this is a complex topic. We have tried to capture the key points you are suggesting briefly in the relevant paragraph of the Discussion by adding (the underlined text):

“...only a small gap. Taiwan lacks survey data (eg capture-recapture studies) to directly inform on the magnitude of this gap. In settings where this gap is larger and changing over time, interpretation of TB notification data is more problematic and notifications may not be a good proxy for incidence. Even in a declining TB epidemic with evolving case-mix, it is possible that case detection may change differently over time in different age-groups; we have not attempted to include such effects. Taiwan does not...”

-3. In line 224, we read: "age groups below 65 will be gradually decreasing whereas the above 65 will nearly stay constant from 2018 to 2035"

It took me some time to understand that the age-specific incidence is proportional to the area under the curves, but not to the lines (i.e. that the histograms are stacked), this probably should be stated more clearly. Also, and more important, in figures 3C-3D, four shades are included in the legend, but only three can be appreciated in the figures.

Thanks for noting these potential sources of confusion. To help readers interpret the meaning of the histogram correctly, we have changed the sentence around line 224 introducing Fig 3C to read:

“Fig 3C shows the overall incidence rates by age group as a stacked histogram.”

To avoid confusion if readers can’t make out the color for the youngest age group, we have changed the last sentence of this section to read:

“In 2035, more than 97% of incident cases will occur among those aged 35 years or older, indeed the contribution from cases under 15 years is nearly invisible in Figures 3C and 3D.”

4. The text is very well written, I only found the following couple of typos:

Line 195: "although it cost a higher degree of freedom" should read "it costs"

Line 262: "Table 2. Summary of reductions in TB incidence reductions with and without

263 demographic change" (remove the second "reductions"?)

Thanks for spotting these typos. We have corrected the errors.

---

## [Decision Letter · Decision Letter 1]

11 Sep 2019

[EXSCINDED]

Forecasting the impact of population ageing on tuberculosis incidence

PONE-D-19-18299R1

Dear Dr. Ku,

We are pleased to inform you that your manuscript has been judged scientifically suitable for publication and will be formally accepted for publication once it complies with all outstanding technical requirements.

With kind regards,

Michele Tizzoni

Academic Editor

PLOS ONE

Additional Editor Comments (optional):

Reviewers' comments:

Reviewer's Responses to Questions

**Comments to the Author**

1. If the authors have adequately addressed your comments raised in a previous round of review and you feel that this manuscript is now acceptable for publication, you may indicate that here to bypass the “Comments to the Author” section, enter your conflict of interest statement in the “Confidential to Editor” section, and submit your "Accept" recommendation.

Reviewer #1: All comments have been addressed

Reviewer #2: (No Response)

2. Is the manuscript technically sound, and do the data support the conclusions?

Reviewer #1: Yes

Reviewer #2: Yes

3. Has the statistical analysis been performed appropriately and rigorously? 

Reviewer #1: Yes

Reviewer #2: Yes

4. Have the authors made all data underlying the findings in their manuscript fully available?

Reviewer #1: Yes

Reviewer #2: Yes

5. Is the manuscript presented in an intelligible fashion and written in standard English?

Reviewer #1: Yes

Reviewer #2: Yes

6. Review Comments to the Author

Reviewer #1: (No Response)

Reviewer #2: The authors have been responsive to my previous comments; I have just caught a couple of confusing sentences which would need some rewording:

-1. In the main text, lines 311-312: "For latent TB, which is accumulated during one's lifetime and depends on historical TB prevalent TB in history"

-In supplementary appendix S1, the first paragraph of section 1.4 contains a number of grammar errors, please revise.

Being these details the only issues I found, I am therefore happy to recommend the work for publication in PLoS One.

7. PLOS authors have the option to publish the peer review history of their article (what does this mean?). If published, this will include your full peer review and any attached files.

Reviewer #1: No

Reviewer #2: No

---

## [Editor Report · Acceptance letter]

16 Sep 2019

PONE-D-19-18299R1 

Forecasting the impact of population ageing on tuberculosis incidence 

Dear Dr. Ku:

I am pleased to inform you that your manuscript has been deemed suitable for publication in PLOS ONE. Congratulations! Your manuscript is now with our production department. 

With kind regards,

on behalf of

Dr. Michele Tizzoni 

Academic Editor

PLOS ONE